# Antioxidant Constituents and Activities of the Pulp with Skin of Korean Tomato Cultivars

**DOI:** 10.3390/molecules27248741

**Published:** 2022-12-09

**Authors:** Dong-Min Kang, Ji-Min Kwon, Woo-Jin Jeong, Yu Jin Jung, Kwon Kyoo Kang, Mi-Jeong Ahn

**Affiliations:** 1College of Pharmacy and Research Institute of Pharmaceutical Sciences, Gyeongsang National University, Jinju 52828, Republic of Korea; 2Division of Horticultural Biotechnology, Hankyong National University, Anseong 17579, Republic of Korea; 3Institute of Genetic Engineering, Hankyong National University, Anseong 17579, Republic of Korea

**Keywords:** tomato, carotenoids, tocopherols, antioxidant activity, correlation analysis

## Abstract

Tomato is a widely distributed, cultivated, and commercialized vegetable crop. It contains antioxidant constituents including lycopene, tocopherols, vitamin C, *γ*-aminobutyric acid, phenols, and flavonoids. This study determined the contents of the antioxidant components and activities of the pulp with skin of ten regular, six medium-sized, and two small cherry tomato cultivars at red ripe (BR + 10) stage cultivated in Korea. The relationships among the Hunter color coordinates, the content of each component, and antioxidant activities were measured by Pearson’s correlation coefficients. As the *a** value increased, the carotenoid and vitamin C contents increased, while the *L** value, hue angle and tocopherol content decreased. As the *b** value increased, the lycopene and total carotenoid contents decreased, and the flavonoid content in the hydrophilic extracts increased. The contents of vitamin C and total carotenoids including lycopene showed high positive correlations with the DPPH radical scavenging activities of both the lipophilic and hydrophilic extracts. Tocopherols and total phenolics in the hydrophilic and lipophilic extracts were not major positive contributors to the antioxidant activity. These findings suggest the quality standards for consumer requirements and inputs for on-going research for the development of better breeds.

## 1. Introduction

Tomato (*Solanum lycopersicum* L.) belongs to the Solanaceae family and is a vegetable crop that originated from South America. It is consumed in various preparations worldwide and contains antioxidants, such as lycopene, *β*-carotene, *γ*-aminobutyric acid (GABA), vitamins C and E, phenolics, and flavonoids [1,2,3].

Carotenoids, which are natural lipophilic pigments of red, yellow, or orange color, are present in various foods and microorganisms. They assist plants in photosynthesis and photoprotection. Lycopene, an abundant antioxidant component of tomato, prevents aging and keeps cells young by removing free radicals in the body [4,5]. It is responsible for the red color of tomatoes and makes up more than 80% of the total carotenoids in red ripe tomato [6,7]. *β*-Carotene acts as provitamin A and inhibits lipid peroxidation induced by xanthine oxidase [8]. Lutein, a xanthophyll carotenoid, has antioxidant activity that scavenges reactive oxygen species (ROS) and reduces retinal oxidative damage [9]. Tocopherols with isoprenoid units are liposoluble antioxidants and a major form of vitamin-E-containing tocotrienols. They inhibit lipid peroxidation and prevent cardiovascular diseases [10]. A previous study showed significant synergistic antioxidant activity of the α-tocopherol-lycopene mixture including other combinations of α-tocopherol and carotenoids [11].

GABA is a non-protein amino acid composed of four carbon atoms. It is produced by the decarboxylation of l-glutamate and l-glutamate decarboxylase (GAD). GABA increases the levels of acetylcholine as a neurotransmitter and has a physiological activity of promoting brain metabolism. In addition, it has many functions such as antioxidant activity, regulation of growth hormone secretion, pain relief, and lowering blood pressure [5,12].

Vitamin C, a hydrophilic water-soluble compound, is another abundant antioxidant constituent in tomatoes found along with lycopene [13]. l-ascorbic acid is a more active dietary form of vitamin C than dehydroascorbic acid, which is the oxidized form. Vitamin C is essential for the biosynthesis of collagen, carnitine, and other neurotransmitters. It protects humans against oxidation of low-density lipoproteins by different oxidative stresses [13].

Accumulation of lycopene in tomato accelerates from the pink stage, and its content is highest at the red ripe stage [14]. The most important external criterion for assessing ripeness of a tomato is its color, which is directly related to the quality perception [15]. In previous studies, tomatoes ripened on the vine showed significantly higher lycopene and *β*-carotene contents than those ripened off vine [16]. In addition, significant differences were found between the metabolite composition of on- and off-vine ripened tomatoes [17]. In the commercial market, vine-ripened tomatoes cost more than the postharvest ripened ones. Tomato fruit can be divided into three parts: skin, pulp, and juice with seeds. Although the skin and seed fractions are commonly discarded during the processing of tomatoes into paste or home-processing methods, the skin is known to have higher levels of lycopene, total phenolics and flavonoids, ascorbic acid, and other antioxidants than pulp and seed fractions. The seed fraction has the lowest amounts of antioxidant constituents [1,6,18].

In this study, we investigated the pulp with skin of ten regular, six medium-sized, and two small yellow cherry tomato cultivars cultivated in South Korea and harvested at red, BR + 10 stage (10 days after breaker stage) based on colorimetric value, total soluble solids (TSS), and antioxidant contents. The 2,2-diphenyl-1-picrylhydrazyl (DPPH) radical scavenging activity of each extract was measured, and the relationships among the Hunter color coordinates, antioxidant chemical constituents, and antioxidant activities were observed.

## 2. Results

### 2.1. Physochemical Properties

Uniformly ripe healthy tomato fruits were harvested from 18 Korean cultivars (Figure 1). The mean fruit weight of the regular cultivars ranged between 180 and 250 g. The six medium-sized and two small cherry tomato cultivars weighed from 50 to 61 g and 20 g, respectively. The firmness of ten regular tomatoes at red ripe stage (reaching full red, BR + 10 stage) was in the range of 2.00–2.80 N (Table 1), which was lower (8.31–11.88 N) than those of the four Korean regular cultivars at pink stage [2]. However, these values are above the recommended limits for home consumption (1.28 N) and retailers (1.46 N) [19]. Medium-sized and small cherry tomatoes showed significantly higher firmness (3.00–4.08 N) than regular tomatoes. No significant difference in firmness was observed between the medium-sized and small cherry tomatoes.

Total soluble solids (TSS) and acidity of fresh fruit affect the sweetness, sourness, and overall quality of flavor [20]. The TSS values of ten regular Korean cultivars at red ripe stage were in the range of 4.70–6.84 °Brix, which were higher compared to those of seven Korean cultivars at pink stage, with values ranging 4.10–5.13 °Brix [21]. Two small yellow cherry cultivars at red ripe stage also showed higher TSS values of 11.24 and 12.34 °Brix, respectively, than those recorded in 13 small cherry cultivars at pink stage (6.07–8.77 °Brix) [21]. Six medium-sized fruits showed TSS values similar to those of the small cherry tomatoes. The TSS values of medium-sized and small cherry tomatoes were much higher than those of regular tomatoes.

The mean total acid contents in the pulp with skin of ten regular cultivars ranged between 0.35 and 0.60 mg CAE (citric acid equivalent)/10 g dry weight. These values were similar to those of fresh fruits from the four Korean cultivars at the pink stage. Acidity in tomatoes increases during development, reaches a maximum at the breaker stage, and then decreases with further ripening [22]. Among the ten regular tomato cultivars evaluated, the highest total acid was registered in the Datlos, whereas it was lowest in the Happiness. The more significant differences were observed among the medium-sized cultivars. The exceptionally high mean value of 0.64 mg CAE/10 g was determined in Blackchoco, while the lowest was observed in Tamina with the value of 0.30 mg CAE/10 g. The remaining five medium-sized tomatoes displayed lower total acid than the regular ones. The acid content was also lower in the two small cherry tomatoes compared to those of the regular and medium-sized cultivars.

The Brix/acid ratio (BAR) affects the taste of tomato juice in terms of sharpness and blandness [2]. In this study, the mean BARs were in the range of 9.26–17.54 and 15.44–34.33 for regular and medium-sized groups, respectively. Thus, medium-sized tomatoes exhibited higher values than those of regular tomatoes. Super toterang and Tamina, respectively, had the highest values in each group. The two small yellow cherry tomatoes showed the highest mean values of 38.62 and 36.18, respectively.

### 2.2. Colorimetric Evaluation

Carotenoids are lipophilic pigments and a major determinant of fruit color. Lycopene with red color and *β*-carotene with orange color are the major carotenoids in tomatoes. Most carotenoid constituents are present in the pulp and skin. Additionally, the ripeness of tomatoes is easily determined by pulp color, a critical factor for consumer preference along with firmness [14]. Therefore, a colorimetric evaluation was performed to reveal the relationship between the colors of the tomato pulp with skin and carotenoid content (Table 2). The *L** values ranged between 43.2 and 77.1. The highest *L** values were observed in light green colored Greengana and yellow colored Norangdotori, GC-9, and GC-19 cultivars (Figure 1). The *a** value was measured according to redness level of tomatoes. Greengana and Norangdotori exhibited negative *a** values. GC-9, GC-19 (yellow color), and Blackchoco (dark red) exhibited the second lower value near zero. Dafnis and Orange with orange and yellow colors, and Blackchoi with dark red color ranked next, with mean values of 17.4, 11.6, and 18.8, respectively. The *a** values of the other cultivars with pink and red colors ranged between 21.2 and 34.1. TY Altorang, Norangdotori, Orange, GC-9, and GC-19 showed higher *b** values due to the yellow color than the others, with mean *b** values between 21.9 and 33.4. The hue angle, which describes the relative amount of redness and yellowness, is often used as an indicator of tomato color change [14]. A hue angle of 90° indicates a yellow color, while 45° and zero degrees show orange-red and red, respectively [23]. The hue angle of yellow colored Greengana and Norangdotori ranked the first and second with mean vales of 65.1° and 48.1°, respectively, followed by two small yellow cherry tomatoes at 43.2° and 42.4°, respectively. Ten regular tomatoes displayed relatively low mean angles between 13.1 and 23.9°. Datlos recorded the lowest hue, while Dafnis with an orange-yellow color recorded the highest hue.

### 2.3. Antioxidant Constituents in Tomato Pulp with Skin

Tomato is used as a commercial health food. It is known for its antioxidant properties and contains lycopene, tocopherol, vitamin C, and *γ*-amino butyric acid [1,4,24]. Therefore, the antioxidant contents in 18 types of tomatoes were estimated, and statistical analysis showed significant differences among the cultivars (Table 2).

Carotenoids and tocopherols were analyzed using a lipophilic extract prepared by sonication of dried pulp with skin in acetone. Acetone is an efficient solvent for the extraction of lipophilic compounds, such as carotenoids and tocopherols [25]. All-*trans*-*β*-carotene and all-*trans*-lycopene were the major carotenoids with peaks appearing in the same order on the liquid chromatography (LC) chromatograms (Figure 2, Appendix A). No other 9-*cis* or 13-*cis β*-carotene isomers were detected in any of the samples. A *cis* isomer of lycopene, 5*Z*-lycopene and lutein were also detected. Lycopene contents were determined by the sum of peak areas of all-*trans*- and 5*Z*-lycopene. The lycopene was the highest among the three carotenoids, and eighteen cultivars evaluated showed significantly different carotenoid contents. The highest lycopene content was found in Datlos and Pinktop with 445.2 and 410.6 μg/g dry weight, respectively. The mean values for the other regular tomatoes were between 173.6 and 360.1 µg/g. Among the medium-sized tomatoes, Blackchoi had the highest lycopene content of 294.5 μg/g. Lycopene was not determined in Greengana, Norangdotori, Orange, GC-9, or GC-19, while the mean values of lycopene contents were 190.0 and 246.4 μg/g for Blackchoco and Tamina, respectively. The *β*-carotene in regular tomatoes ranged from one-third to one-sixteenth of the lycopene content, with mean values between 18.1 and 49.6 μg/g. The *β*-carotene ranged from 36.5 to 55.0 μg/g in Blackchoi, Blackchoco, Orange, and Tamina, while in the other cultivars it was between 0.9 and 2.6 μg/g. Lutein content was the lowest among the three carotenoids in all the cultivars. The mean values were between 2.9 and 8.9 μg/g, except Blackchoco with highest value of 26.2 μg/g. The total carotenoid content was proportional to the lycopene content with mean values ranging from 348.9 to 674.4 μg/g, except for five cultivars, namely, Greengana, Norangdotori, Orange, GC-9, and GC-19, in which lycopene was not detected and the values were only between 6.6 and 71.2 μg/g.

*α*-, *γ*-, and *δ*-tocopherols are the major types of tocopherols present in the 18 tomato cultivars. *α*-Tocopherol was the most abundant among the three, followed by *γ*- and *δ*-tocopherols. The two small yellow cherry tomatoes showed the highest *α*-tocopherol (61.8 and 58.3 μg/g) and total tocopherol contents (65.9 and 63.3 μg/g). The mean values of *α*-, *γ*- and *δ*-tocopherol contents in the other cultivars were in the range of 14.6–30.2, 0.48–10.1, and 0.11–1.65 μg/g, respectively, except for Tamina, which showed the lowest α-tocopherol (5.5 μg/g) and total tocopherol (10.2 μg/g) contents.

Although devoid of any antioxidant constituents, the content of chlorophylls—the green photosynthetic pigment of plants—was also determined using the lipophilic extract under the same LC conditions used for carotenoid analysis. Chlorophylls a and b were not detected in ten regular and one medium-sized (Tamina) tomato cultivars at the red ripe stage. Blackchoi and Blackchoco showed significantly similar higher contents of chlorophyll a (65.6 and 79.8 µg/g, respectively). Their values were followed by those of Greengana and Orange, ranked the second and the third highest, with mean values of 50.5 and 23.8 µg/g, respectively. GC-9 and GC-19 showed similar low levels of chlorophyll a and b (5.9–8.3 µg/g). Blackchoco registered the highest chlorophyll b (56.1 µg/g), while the mean values in the other medium-sized cultivars were between 9.8 and 27.5 µg/g—except for Tamina, which failed to display chlorophyll.

The free amino acids contents, including that of GABA, were determined using 70% ethanol extract. l-Glutamate, glutamine, aspartic acid, asparagine, and GABA were the major free amino acids in the tomatoes, and these findings are similar to those from previous reports (data not shown) [2,26,27]. The mean values of GABA were between 2.1 and 12.3 mg/g dry weight. Two regular tomato cultivars, Datlos and Pinktop, contained relatively high GABA of 11.8 and 12.3 mg/g, respectively. Three regular tomato cultivars, Dongyu250 ho, Happiness, and Madison, and two medium-sized cultivars, Blackchoco and Greengana, ranked the second high with mean values between 6.1 and 8.2 mg/g. The mean values of total amino acid in 18 cultivars ranged from 24.9 to 47.5 mg/g.

Vitamin C content was determined as ascorbic acid equivalent (AAE) using 3% metaphosphoric acid extracts. While Greengana, Norangdotori, Orange, GC-9, and GC-19 showed much lower AAE levels in the range of 0.23–1.10 mg AAE/g, the mean values of the other cultivars lied between 1.70 and 2.90 mg AAE/g.

### 2.4. Antioxidant Activity of Lipophilic and Hydrophilic Extracts

Phenolics act as natural antioxidants in plants and prevent excess free-radical-related diseases by reducing oxidative stress [28,29]. Flavonoids are phenolics known to have antioxidant activities significantly contributing to the health benefits of tomatoes [13]. Therefore, the total phenolic and flavonoid content can indirectly confirm antioxidant activity. In this study, the total phenolic and flavonoid content was determined to investigate the effects of these compounds on the antioxidant activities of lipophilic and hydrophilic extracts, respectively. Lipophilic extracts are rich in carotenoids and tocopherols and are prepared from lyophilized tomato pulp (with skin) with acetone solvent. Hydrophilic extracts are rich in ascorbic acid and amino acids, including GABA, prepared with methanol solvent by ultrasonication.

As per results, total phenolic contents in lipophilic and hydrophilic extracts were similar to each other, and the mean values were between 12.8 and 21.4 µmol gallic acid equivalent (GAE)/g, except for TY Altorang and Tamina. The total phenolic contents in the hydrophilic extracts of TY Altorang and Tamina were 27.2 ± 1.3 and 20.3 ± 0.2 µmol GAE/g, respectively, which are 1.4- and 1.5-fold higher than those in the lipophilic extracts, respectively. The mean values of total flavonoid content in the hydrophilic extract were in the range of 5.5–10.8 µmol QE/g, while flavonoids were not detected in all lipophilic extracts (Table 2). Norangdotori, Orange, GC-9, and GC-19 with yellow color showed higher flavonoid contents than the other cultivars.

Antioxidant activity was measured by DPPH radical scavenging assay using lipophilic and hydrophilic extracts (Table 2). The DPPH radical scavenging activity of hydrophilic extracts was 1.2–3.2 fold higher than that of lipophilic extracts, except for Happiness, in which the antioxidant activities of lipophilic and hydrophilic extracts were similar each other with values of 27.3 ± 3.5 and 27.4 ± 1.8 µmol triolox equivalent (TE)/g, respectively. The mean values of lipophilic extracts were between 8.8–29.7 µmol TE/g. The values for hydrophilic extracts were in the range of 15.7–41.9 µmol TE/g. The antioxidant activities of cultivars with low lycopene or vitamin C contents were lower than those of other cultivars. A medium-sized tomato cultivar, Tamina, showed the strongest DPPH radical scavenging activity for both lipophilic and hydrophilic extracts.

### 2.5. Correlations among Physicochemical Data, Antioxidant Constituent, and Antioxidant Activity

To reveal the correlation among the physicochemical data, antioxidant constituent contents, and antioxidant activity, Pearson’s correlation coefficients were obtained by performing a correlation analysis based on data from ten regular tomato and six medium-sized tomato groups, respectively, (Table 3 and Table 4), followed by the analysis of all 18 cultivars (Table 5). Firmness, TSS, and BAR were excluded from the correlation analysis for all 18 samples, because these elements showed significant differences among the regular, medium-sized, and small cherry tomato groups.

In ten regular tomatoes at the red ripe stage, the content of total soluble solids (Brix) of total fruit exhibited significant positive correlations with total phenolic content of the lipophilic extracts from tomato pulp with skin, while it showed negative correlation with GABA content. Brix displayed no significant correlation with any other factor in the six medium-sized tomatoes at the red ripe stage. Total acids showed significant positive correlation with GABA and total amino acid content in regular tomatoes, while it showed a significantly high positive correlation with lutein content and a weak positive correlation with GABA content in medium-sized tomatoes.

As the brightness (*L** value) increased, the hue angle (*h*°) increased and the redness (*a** value) decreased in both regular and medium-sized tomatoes. Meanwhile, the *L** value was negatively correlated with lycopene and total carotenoid content in regular tomatoes, whereas it showed a positive correlation with total tocopherol content. In medium-sized tomatoes, the *L** value was negatively correlated with *β*-carotene content. As the *a** value increased, the hue angle decreased in both regular and medium-sized tomatoes. In regular tomatoes, the *a** value showed positive correlations with lycopene and total carotenoid contents, while it showed a negative correlation with total tocopherol content. In medium-sized tomatoes, the *a** value exhibited a significantly positive correlation with the DPPH radical scavenging activity of the lipophilic extracts. The yellowness (*b** value) displayed no significant correlation with any other factor in either regular or medium-sized tomatoes. The hue angle was negatively correlated with lycopene and total carotenoid contents in both regular and medium-sized tomatoes. In addition, hue angle also showed a negative correlation with GABA content in regular tomatoes, while it showed a negative correlation with DPPH radical scavenging activity in hydrophilic extracts of medium-sized tomatoes.

The lutein content exhibited no significant correlation with any other element in any of the 18 samples. Since lycopene is the most abundant carotenoid in tomatoes, a significantly high positive correlation (*r* = 0.99) between lycopene content and total carotenoid content in all cultivars was observed, whereas *β*-carotene content showed a weak positive correlation (*r* = 0.56) with total carotenoid content. In addition, lycopene and total carotenoid contents were positively correlated with GABA and ascorbic acid contents, and DPPH radical scavenging activities of lipophilic and hydrophilic extracts in all cultivars. However, they showed significantly negative correlations with the total tocopherol content and total flavonoid content in the hydrophilic extracts.

In contrast, total tocopherol content was negatively correlated with DPPH radical scavenging activities of lipophilic and hydrophilic extracts in all cultivars, whereas it had a positive correlation with total flavonoid content in hydrophilic extracts. GABA content was negatively correlated with the total phenolic content in lipophilic extracts and the total flavonoid content in hydrophilic extracts. The total acid content exhibited no significant correlation with any other elements in any of the samples. In particular, there was a significantly high positive correlation (*r* = 0.91) between the ascorbic acid content and DPPH radical scavenging activity of the hydrophilic extracts. Ascorbic acid also exhibited a positive correlation with the DPPH radical scavenging activity of lipophilic extracts, while it showed a negative correlation with the total flavonoid content in hydrophilic extracts. The DPPH radical scavenging activity of the lipophilic extracts was negatively correlated with the total flavonoid content in the hydrophilic extracts, whereas it showed a positive correlation with DPPH radical scavenging activity of the hydrophilic extracts. The total phenolic content was not significantly correlated with any other element in any of the cultivars. In particular, the total flavonoid content in hydrophilic extracts was negatively correlated with the DPPH radical scavenging activity of both lipophilic and hydrophilic extracts.

## 3. Discussion

Tomato is a globally distributed, cultivated, and commercialized vegetable. It contains antioxidants such as lycopene, tocopherols, vitamin C, *γ*-aminobutyric acid, phenols, and flavonoids. This study determined the contents of the antioxidants and their antioxidant activities in the pulp with skin of tomato fruits ripened on the vine from ten regular, six medium-sized, and two yellow small cherry cultivars cultivated in Korea. Statistical analysis showed significant differences in the contents among the cultivars. Correlation analysis was performed using Pearson’s correlation coefficients to identify significant interactions among the studied variables of Hunter color coordinates, content of each antioxidant constituents, and antioxidant activities.

It is known that Brix, *β*-carotene, lycopene, and vitamin C contents increase, and the GABA content decreases, as tomato ripens and turns from green to red [14,30,31]. The BAR is higher in mature fruits than in immature fruits [32]. In addition, vine-ripening increased the soluble solids of the fruit more than chamber-ripening, while the acid levels of fruits that underwent these two different ripening processes were totally opposite. The chamber-ripened fruits had the higher acid content. This difference in the sugar–acid ratio between the two ripening processes is likely associated with different rates of respiratory breakdown of sugars and organic acids in the fruits [32]. At the red ripe stage, the firmness was lower and TSS content and BAR were higher than those previously reported for tomatoes at the pink stage [2,21]. The present results in this study are consistent with these previous reports and suggest that the firmness has negative correlations with TSS content and BAR.

The distinctive red color of tomato fruit is due to the accumulation of lycopene [6,7,33] and this resulted in the high correlation (*r* = 0.99) between lycopene content and total carotenoid contents in the present study. The lycopene accumulation takes place during ripening from the breakdown of chlorophylls as a result of genetic programming, and consequently, the greenness disappears [34]. In this study, the *a** values of redness were higher in eight regular tomato cultivars at the red ripe stage than those of the four Korean tomato cultivars at the pink stage as per a previous report [2]. Blackchoco, a medium-sized cultivar, registered exceptionally high lutein content among all samples, although lycopene content was lower compared to another medium-sized cultivar, Blackchoi, both with a similar color. Hence, Blackchoco merits promotion for better vision. Blackchoco also exhibited the highest *β*-carotene content among all samples with another medium-sized cultivar, Blackchoi, and a regular tomato, Dafnis, although *b** values (yellowness) of Blackchoco were relatively low.

GABA accumulates in tomatoes before the breaker stage and catabolizes rapidly thereafter [35]. Therefore, the GABA content was significantly reduced during the ripening transition [27]. In this study, the GABA contents of sixteen cultivars except Datlos and Pinktop, at red ripe stage, were below the level expressed by the “Micro-Tom” cultivar at the breaker stage [35].

The *a** value of redness had a high correlation (*r* = 0.84) with lycopene in all samples, indicating that as lycopene increases, the color of tomatoes become more red. The hue angle, which describes the relative amounts of redness and yellowness, showed higher negative correlations with lycopene and total carotenoid (*r* = −0.85 and −0.87, respectively) than the positive correlations (*r* = 0.84 and 0.85, respectively) of *a** values with these contents. The difference in the correlation between the *a** value and hue angle was larger in six medium-sized cultivars than in ten regular cultivars. Moreover, a negative correlation existed between the hue angle and vitamin C. These results were consistent with those of previous reports that hue angle could be used as a good indicator of the functional quality of tomatoes [14,21]. However, in contrast to the previous report, no correlation was observed between the *a** value or hue angle and total amino acid contents in our study. This is possibly because the amino acid content in pulp with skin utilized in this study may not be dominant for total amino acid content as observed for whole tomato fruit. The lycopene and vitamin C contents in the pulp with skin were much higher than those in the seed fraction [1]. As the *a** value increased and hue angle decreased, the carotenoid and vitamin C increased, while the *L** value and tocopherol content decreased.

The yellowness *b** value showed a high positive correlation (*r* = 0.80) with the total flavonoid content in the hydrophilic extract. As the *b** value increased, while the lycopene and total carotenoid contents decreased, the total flavonoid content in the hydrophilic extracts increased. This strong correlation suggests that the yellowness of tomatoes appears from yellow flavonoids. In fact, four yellow cultivars, Norangdotori, Orange, GC-9, and GC-19 with high *b** values showed higher total flavonoids in hydrophilic extracts than other cultivars. In ten regular tomato cultivars with relatively lower total flavonoids than medium-sized and small cherry cultivars, the highest Hunter *b** value was recorded in TY Altorang, which also had the highest total flavonoid among the regular tomatoes. No significant correlation between the *b** value and the content of another yellow lutein or *β*-carotene was observed. This result is different from a previous report demonstrating more yellowness due to the higher content of *β*-carotene [36]. This discrepancy could be ascribed to the fact that only one tomato cultivar at three ripening stages was studied in the previous study, while our study sampled 18 cultivars at the same ripening stage [36]. Exceptionally, Orange cultivar with orange color displayed a relatively higher *β*-carotene content than others. Lycopene was not detected in the five cultivars of Greengana, Norangdotori, Orange, GC-9, and GC-19. Norangdotori, Orange, GC-9, and GC-19 with yellow or orange color possessed higher flavonoid content than the other cultivars. Therefore, the orange color of the Orange cultivar can be attributed to the high *β*-carotene and flavonoid contents. These five cultivars also showed much lower vitamin C levels than other cultivars. The green color of Greengana can be attributed to its high chlorophyll content. While chlorophylls a and b were not detected in any of the ten regular tomatoes at the red ripe stage, they were detected in five medium-sized (including Orange cultivar) and two small yellow cherry tomatoes.

The contents of vitamin C (ascorbic acid) and total carotenoids, including lycopene, showed high positive correlations with the DPPH radical scavenging activities of both the lipophilic and hydrophilic extracts. This result is in line with that of a previous report [37]. In particular, a significantly high positive correlation (*r* = 0.91) between the ascorbic acid content and DPPH radical scavenging activities was observed in the hydrophilic extract. This result could be ascribed to the fact that, with respect to contents and antioxidant capacity, vitamin C is the major antioxidant constituent in hydrophilic extracts while lipophilic extracts are dominated by carotenoids. Among the five antioxidants evaluated, the trolox equivalent antioxidant capacity decreases in the following order: lycopene (2.2–3.1), *β*-carotene (1.5–2.0), vitamin C (1.1), α-tocopherol (0.9–1.0), and GABA (0.3–0.7) compared to trolox (1) in ABTS or DPPH assay [38,39,40]. In this study, vitamin C levels were found to increase as the contents of total carotenoids including lycopene increased.

Lycopene content did not show any significant correlation with antioxidant activities assayed by DPPH or ABTS radical scavenging as reported in previous studies [18,37]. However, in this study, lycopene and total carotenoid contents showed the same high positive correlation (*r* = 0.81) with DPPH radical scavenging activities of lipophilic extracts from tomato pulp with skin at the red ripe stage. This discrepancy could be due to extraction of lipophilic substances by acetone solvent for measuring antioxidant activity instead of methanol that was used in the previous study. This also means that lipophilic extract by acetone solvent could better reflect the content of lycopene, a lipophilic constituent, than the methanol extract or other hydrophilic extracts. In addition, lycopene and total carotenoid contents also showed significant positive correlations (*r* = 0.76 and 0.79, respectively) with the antioxidant activities of hydrophilic extracts. This correlation could be due to the positive relationship between these contents and the major antioxidants (vitamin C and GABA) in hydrophilic extracts. Consequently, as the DPPH radical scavenging activity of the lipophilic extracts increased, the activity of the hydrophilic extracts also increased.

Hydrophilic extracts showed higher DPPH radical scavenging activity than lipophilic extracts. This may be derived from the fact that the contents of major antioxidant constituents including vitamin C and GABA in hydrophilic extracts were much higher than those of antioxidants, including total carotenoids and tocopherols in the lipophilic extracts. Fourteen small cherry tomato cultivars and four regular cultivars also showed higher antioxidant activities in the hydrophilic extracts than in the lipophilic extracts [41]. This result was also observed in lipophilic and hydrophilic extracts from skin, pulp, and seeds of tomato [18]. The total contents of phenolics, yet another source of antioxidant, in hydrophilic extracts failed to show a significant positive correlation with antioxidant activity. This result is consistent with that of a previous study [41]. The total phenolic content in lipophilic extracts also failed to show a significant positive correlation with the antioxidant activity of the extracts.

Vitamin C content was negatively correlated with total flavonoid content in the hydrophilic extracts. Flavonoid was not detected in the lipophilic extracts. The total flavonoid content in the hydrophilic extract showed positive correlation with tocopherol content and *b** value, whereas it was negatively correlated with lycopene, total carotenoid, and vitamin C contents.

In conclusion, this study revealed that the color attributes of tomato pulp with skin, redness, and hue angle would be factors to assume the contents of lycopene, total carotenoids, and vitamin C, with yellowness for the total flavonoid content. As lycopene content increased, the vitamin C and GABA contents and antioxidant activities of lipophilic and hydrophilic extracts increased. Tocopherols and total phenolics in the hydrophilic and lipophilic extracts were not major positive contributors to the antioxidant activity. Although studies on the antioxidant activity and components of tomato cultivars have been reported, no study has investigated the antioxidant activity of lipophilic extracts containing carotenoid constituents or tomato pulp with skin at the red ripe stage from tomato cultivars cultivated in Korea [2,10,13]. These findings provide data to fix quality standards for consumer requirements and preferences and provide inputs for ongoing research for the development of a more competitive new breed of tomato. For instance, a more red-colored tomato has a higher possibility of containing high levels of carotenoids, GABA, and vitamin C. In addition, if the target is lycopene or GABA, then two regular tomatoes, Datlos and Pinktop, are the best choice. On the other hand, if the target is vitamin C, then a regular Dongyu 250 ho and a middle-sized Tamina are the best choice.

## 4. Materials and Methods

### 4.1. Plant Materials

Eighteen tomato cultivars of ten regular, six medium-size, and two cherry cultivars cultivated in Korea were used in this study (Table 1). The seeds were sown on plug trays on 5 April 2021, and 35-day-old seedlings were transplanted to a greenhouse at the Hankung National University, Korea, with a planting distance of 50 × 90 cm. The fertigation solution was prepared with EC 2.0–2.2 dS m*^−^*^1^ and pH 5.5–5.8 according to the growth stages. No incidence of disease was observed during the cultivation period. Simple pruning and training the plant for straight growth with nylon strips were carried out every day. Mature fruits of 1.5 kg at red ripe stage (BR + 10 stage) were harvested on 5 August 2021 from each cultivar for physicochemical and phytochemical analysis. Color attributes were evaluated within 6 h of harvest. The fruits were free from any physical defects, and of uniform size at the red ripe stage, which was determined and selected under the United States Department of Agriculture (USDA) tomato ripeness color classification chart [42]. Fruit firmness was measured using a Compac-10011 rheometer (Sun Scientific Co., Ltd., Tokyo, Japan). The physicochemical data were taken immediately after the selection, and the pulp with skin was taken from the other jelly and juicy parts. The pulp with skin of each cultivar was pulverized with liquid nitrogen gas and freeze-dried and kept in –80 °C until phytochemical analysis and antioxidant assay.

### 4.2. Total Soluble Solids (TSS), Total Acids (TA) of Pulp, and Brix Acid Ratio (BAR)

The content of total soluble solids was determined from five fresh fruit samples at room temperature using an Atago DR-A1 digital refractometer (Atago Co., Ltd., Tokyo, Japan) by the previously reported method [2]. TA content was determined by titration of water extract of each sample with 0.1 N NaOH solution as citric acid equivalents per gram (μg CAE/g). Briefly, one hundred milligrams of lyophilized tomato pulp with skin were extracted with 10 mL of distilled water by sonication three times for 10 min each. The extract was centrifuged at 5700× *g* at room temperature for 10 min, and supernatant was filtered with a 0.45 μm filter. The pH value of the filtrate with the volume of 5 mL was adjusted with 0.1 N NaOH solution to pH 8.1 using a pH meter (Orion Star™ A211 pH Benchtop Meter, ThermoFisher, Waltham, MA, USA). BAR was obtained by dividing the TSS value by the TA value.

### 4.3. Colorimetric Evaluation

The color of the tomato was evaluated using a colorimeter (CR-400 Chroma Meter, Konica Minolta, Inc., Osaka, Japan). Color correction was carried out using standard colors. The Hunter color coordinates were expressed as brightness (*L**), redness (*a**), and yellowness (*b**) values, respectively. Hue angle was obtained from the formula, *h*° = tan^−1^ (*b**/*a**).

### 4.4. Analysis of Carotenoids and Tocopherols

The lipophilic extract was used for the analysis of carotenoids and tocopherols. It was prepared from the pulp with skin of tomato fruits with acetone (0.01% butylated hydroxytoluene, BHT) by our previously reported methods [25,43]. HPLC analysis was performed according to our previously reported method using Agilent 1260 HPLC system with some modification on gradient system of eluent solution (Hewlett-Packard, Waldbronn, Germany). Solvents were HPLC grade of water, methanol, and methyl-*tert*-butyl ether (MTBE). The solvent mixture consisted of methanol:MTBE:water (81:15:4; A) and methanol:MTBE:water (6:90:4; B). The chromatogram was performed by a gradient elution under the following conditions: 0 to 15 min, 0% B; 15 to 50 min, 100% B; 50 to 60 min, 100% B. The flow rate was 0.7 mL/min, and the temperature was maintained at 30 °C. The peaks of carotenoid standards such as lutein, all*-trans-β*-carotene, and all-*trans*-lycopene from Carotenature (Münsingen, Switzerland) were measured at 450 nm absorbance. Chlorophylls a and b (Sigma-Aldrich) was quantified using the same LC conditions. Under these conditions, each standard peak eluted at the following *t_R_* (min): 11.1 for violaxanthin, 18.8 for chlorophyll b, 20.9 for lutein, 25.8 for zeaxanthin, 26.8 for chlorophyll a, 35.7 for 13 *Z*-*β*-carotene, 36.5 for *α*-carotene, 38.4 for all-*trans*-*β*-carotene, 39.6 for 9 *Z*-*β*-carotene, 52.6 for all-*trans*-lycopene, and 53.3 for 5 *Z*-lycopene.

Tocopherol analysis was accomplished by the previously reported method with slight modifications [44]. Briefly, the same HPLC system used for carotenoid analysis was used with the same eluent solutions. The chromatogram was performed by a gradient elution under the following conditions: 0 to 16 min, 0% B; 16 to 20 min, 100% B; 20 to 30 min, 100% B. The flow rate was 0.7 mL/min, and the temperature was maintained at 30 °C. Peak detection was measured by fluorescence (*Ex.* 298 nm and *Em.* 325 nm). Under these conditions, standard tocopherol peaks (Sigma-Aldrich, St, Louis, MO, USA) eluted at the following *t_R_* (min): 10.1 for *δ*-tocopherol, 11.5 for *γ*-tocopherol, 12.2 for *β*-tocopherol, and 13.3 for *α*-tocopherol.

### 4.5. Analysis of GABA and Free Amino Acids

The contents of free amino acids including GABA were determined according to a previously reported method [6]. Two hundred and fifty milligrams of lyophilized tomato was extracted with 10 mL of 70% ethanol by three-times sonication for 10 min each. The extract was centrifuged at 5700× *g* at room temperature for 10 min, and one milliliter of supernatant was filtered with a 0.22 μm filter. The filtrate was analyzed with a Biochrom 30+ Amino Acid Analyzer system (Biochrom Ltd., Cambridge, UK) using a Lithium accelerated Resin H-1649 column (Biochrom Ltd., Cambridge, UK). Absorbance was measured at 570 nm.

### 4.6. Vitamin C Content

The content of vitamin C was determined according to a previously reported method [37]. Two hundred fifty milligrams of lyophilized tomato pulp with skin was extracted with 3% metaphosphoric acid by sonication three times for 10 min each. The extract was centrifuged at 5700× *g* at 4 °C for 10 min, and supernatant was filtered with a 0.45 μm filter. HPLC analysis was performed on Agilent 1260 HPLC system and YMC ODS C18 column (4.6 × 250 mm, 4 μm) with 0.2 M KH_2_PO_4_ (pH 2.2; solvent A) and methanol (solvent B). The gradient elution was applied as follows: 0 to 15 min, 0% B; 15 to 23 min, 100% B; 23 to 24 min, 0% B; 24 to 25 min, 0% B. The flow rate was 0.7 mL/min and injection volume was 20 µL. Column temperature was 30 °C, and absorbance were measured at 254 nm.

### 4.7. Total Phenolic and Total Flavonoid Contents

The lipophilic and hydrophilic extracts were used for the determination of total phenolic and total flavonoid contents, respectively. The hydrophilic extract was prepared with 250 mg of dried pulp with skin and 10 mL of methanol by three-times sonication for 10 min each. The extract was centrifuged at 5700× *g* at 4 °C for 10 min, and supernatant was filtered with a 0.45 μm filter. The lipophilic extract was the extract used for the analysis of carotenoids and tocopherols.

Total phenolic content was measured by Folin–Ciocalteu assay [45]. One hundred microliters of each extract was reacted with 500 μL of 10% Folin–Ciocalteau phenol reagent and 400 μL of sodium carbonate at room temperature for 10 min, and centrifuged 3000× *g* for 5 min. The absorbance was measured at 765 nm after transferring 200 μL of the supernatants to 96 wells. The calibration curve was obtained with gallic acid, and the phenolic content was expressed as gallic acid equivalents per gram (μmol GAE/g).

Total flavonoid content was measured by diethylene glycol assay [45]. Twenty microliter of each extract was reacted with 170 μL of 90% diethylene glycol and 10 μL of 4 M NaOH at room temperature. After 10 min, the absorbance was measured at 420 nm. Calibration curve was obtained with quercetin, and the flavonoid content was expressed as quercetin equivalents per gram (μmol QE/g).

### 4.8. Antioxidant Activity Test with DPPH Radical

DPPH radical scavenging activity was measured by our previously reported method [46]. Each sample of 20 μL was added to 180 µL of 0.2 mM 1,1-diphenyl-2-picrylhydrazyl (DPPH) solution in each well of 96-well plate. After 10 min, the absorbance was measured at 520 nm. The antioxidant activity was expressed as trolox (Sigma-Aldrich, St. Louis, MO, USA) equivalents per gram (μmol TE/g).

### 4.9. Statistical Analysis

All data were expressed as means ± standard deviations (SD) of two or triple independent experiments. The statistical differences among the samples by one-way analysis of variance (ANOVA) and Pearson’s correlation coefficients were obtained using SPSS Statistics 24.0 software (IBM, Armonk, NY, USA). Statistical significance level was performed with 5%.

## Figures and Tables

**Figure 1 molecules-27-08741-f001:**
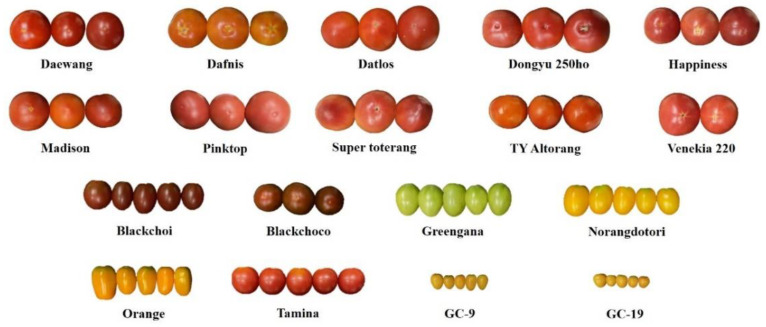
Eighteen tomato cultivars used in this study.

**Figure 2 molecules-27-08741-f002:**
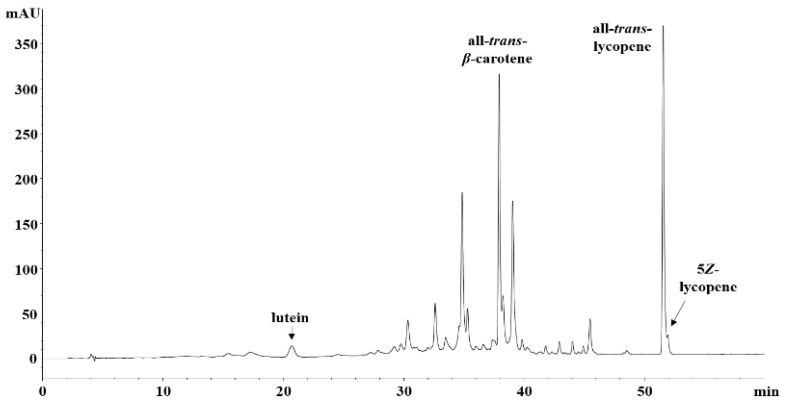
LC chromatogram of the lipophilic extract from a tomato cultivar, Dafnis (450 nm).

**Table 1 molecules-27-08741-t001:** Firmness, TSS, TA, and BAR of eighteen tomato cultivars at red ripe stage.

Cultivar	Category	Firmness (N)	TSS (°Brix)	TA (mg CAE/10 g)	BAR
Daewang	Regular	2.24 ± 0.46 ^b^	5.15 ± 0.21 ^e^	0.39 ± 0.01 ^c^	13.27 ± 0.13 ^e^
Dafnis		2.46 ± 0.46 ^b^	5.80 ± 0.21 ^e^	0.47 ± 0.05 ^b^	12.34 ± 0.78 ^ef^
Datlos		2.12 ± 0.22 ^b^	5.56 ± 0.16 ^e^	0.60 ± 0.01 ^a^	9.26 ± 0.11 ^f^
Dongyu 250 ho		2.00 ± 0.39 ^b^	5.58 ± 0.24 ^e^	0.41 ± 0.02 ^bc^	13.61 ± 0.09 ^e^
Happiness		2.05 ± 0.51 ^b^	5.54 ± 0.18 ^e^	0.35 ± 0.01 ^d^	15.82 ± 0.06 ^d^
Madison		2.17 ± 0.42 ^b^	6.30 ± 0.22 ^d^	0.45 ± 0.02 ^bc^	14.00 ± 0.13 ^e^
Pinktop		2.80 ± 0.21 ^ab^	4.70 ± 0.19 ^f^	0.49 ± 0.03 ^b^	9.59 ± 0.19 ^f^
Super toterang		2.80 ± 0.45 ^ab^	6.84 ± 0.16 ^d^	0.39 ± 0.04 ^c^	17.54 ± 1.26 ^d^
TY Altorang		2.02 ± 0.46 ^b^	6.20 ± 0.16 ^d^	0.41 ± 0.06 ^bc^	15.12 ± 1.59 ^de^
Venekia 220		2.42 ± 0.64 ^b^	6.72 ± 0.29 ^d^	0.43 ± 0.03 ^bc^	15.63 ± 0.39 ^de^
Blackchoi	Medium-sized	4.12 ± 0.27 ^a^	11.2 ± 0.3 ^b^	0.37 ± 0.03 ^cd^	30.27 ± 1.52 ^b^
Blackchoco		4.08 ± 0.31 ^a^	9.88 ± 0.17 ^c^	0.64 ± 0.01 ^a^	15.44 ± 0.02 ^de^
Greengana		3.42 ± 0.46 ^ab^	9.45 ± 0.19 ^c^	0.41 ± 0.04 ^bc^	23.05 ± 1.41 ^c^
Norangdotori		3.46 ± 0.26 ^ab^	10.1 ± 0.2 ^c^	0.37 ± 0.04 ^cd^	27.30 ± 2.18 ^b^
Orange		3.80 ± 0.56 ^a^	11.3 ± 0.3 ^b^	0.38 ± 0.03 ^cd^	29.74 ± 1.45 ^b^
Tamina		3.00 ± 0.89 ^ab^	10.3 ± 0.2 ^c^	0.30 ± 0.02 ^d^	34.33 ± 1.52 ^a^
GC-9	Small Yellow cherry	3.17 ± 0.72 ^ab^	11.2 ± 0.4 ^b^	0.29 ± 0.02 ^d^	38.62 ± 1.20 ^a^
GC-19		3.80 ± 0.32 ^a^	12.3 ± 0.2 ^a^	0.34 ± 0.02 ^d^	36.18 ± 1.46 ^a^

TSS, total soluble solids; TA, total acids; BAR, brix acid ratio. The values represent mean ± SD (*n* = 5). Total acids are expressed as the mean citric acid equivalent (CAE) for dry weight. BAR was determined by dividing the TSS with TA. Different letters in the same column mean significantly different (*p* < 0.05).

**Table 2 molecules-27-08741-t002:** Hunter values and antioxidant constituents in the tomato pulp of eighteen tomato cultivars, and their antioxidant activity.

Items	Cultivars									
	Daewang	Dafnis	Datlos	Dongyu 250 ho	Happiness	Madison	Pinktop	Super Toterang	TY Altorang	Venekia 220
*L**	57.0 ± 0.8 ^bc^	64.9 ± 0.7 ^b^	46.0 ± 1.3 ^de^	62.6 ± 2.4 ^b^	48.9 ± 0.9 ^d^	43.2 ± 2.7 ^e^	53.0 ± 0.9 ^c^	56.3 ± 2.0 ^bc^	54.3 ± 0.6 ^c^	60.9 ± 0.1 ^b^
*a**	27.0 ± 1.2 ^c^	17.4 ± 1.5 ^d^	25.1 ± 0.3 ^c^	21.2 ± 1.1 ^c^	34.1 ± 1.1 ^a^	30.3 ± 0.5 ^b^	31.0 ± 0.7 ^b^	23.2 ± 0.3 ^c^	26.6 ± 0.3 ^c^	22.3 ± 0.1 ^c^
*b**	30.3 ± 1.4 ^c^	28.9 ± 3.6 ^c^	21.9 ± 0.9 ^d^	24.8 ± 1.0 ^d^	33.4 ± 0.5 ^bc^	29.2 ± 0.8 ^c^	28.4 ± 0.7 ^cd^	30.0 ± 0.4 ^c^	38.6 ± 1.6 ^b^	24.7 ±1.1 ^d^
Hue angle (°)	17.6 ± 0.4 ^g^	23.9 ±0.7 ^e^	13.1 ± 0.9 ^h^	18.4 ± 0.3 ^g^	14.6 ± 0.6 ^h^	14.0 ± 0.6 ^h^	13.4 ± 0.7 ^h^	20.5 ± 0.2 ^f^	22.6 ± 0.3 ^e^	17.1 ± 0.8 ^g^
Lutein (µg/g)	8.4 ± 0.2 ^b^	5.4 ± 0.2 ^b^	6.4 ± 0.2 ^b^	7.3 ± 1.9 ^b^	5.3 ± 0.1 ^b^	5.1 ± 0.2 ^b^	6.6 ± 1.2 ^b^	8.1 ± 2.1 ^b^	8.5 ± 0.9 ^b^	4.9 ± 0.7 ^b^
*β*-Carotene (µg/g)	39.3 ± 1.0 ^b^	49.6 ± 3.7 ^a^	28.0 ± 2.5 ^c^	18.1 ± 2.9 ^d^	31.9 ± 0.4 ^bc^	23.3 ± 2.6 ^cd^	24.6 ± 0.9 ^cd^	29.4 ± 2.1 ^c^	26.6 ± 1.6 ^c^	24.0 ± 3.1 ^cd^
Lycopene (µg/g)	272.8 ± 3.4 ^c^	173.6 ± 7.3 ^e^	445.2 ± 24.3 ^a^	225.9 ± 25.6 ^d^	307.0 ± 10.3 ^c^	360.1 ± 16.7 ^b^	410.6 ± 18.4 ^a^	184.8 ± 17.6 ^e^	350.4 ± 34.1 ^b^	229.8 ± 14.8 ^d^
Other carotenoids (µg/g)	163.1 ± 2.1 ^b^	129.0 ± 3.7 ^c^	194.8 ± 21.3 ^a^	142.5 ± 11.0 ^bc^	181.8 ± 1.8 ^ab^	186.0 ± 3.8 ^a^	210.7 ± 10.5 ^a^	126.6 ± 7.4 ^c^	187.0 ± 9.9 ^a^	156.6 ± 6.9 ^b^
**Total carotenoids (µg/g)**	483.6 ± 6.8 ^bc^	357.7 ± 14.9 ^d^	674.4 ± 48.3 ^a^	393.8 ± 41.5 ^d^	526.1 ± 12.7 ^b^	574.5 ± 23.3 ^b^	652.4 ± 30.9 ^a^	348.9 ± 29.2 ^d^	572.5 ± 46.4 ^b^	415.2 ± 25.6 ^c^
Chlorophyll a (µg/g)	ND	ND	ND	ND	ND	ND	ND	ND	ND	ND
Chlorophyll b (µg/g)	ND	ND	ND	ND	ND	ND	ND	ND	ND	ND
*α*-Tocopherol (µg/g)	21.5 ± 0.6 ^c^	30.2 ± 1.2 ^b^	16.5 ± 0.6 ^d^	25.6 ± 0.3 ^b^	14.6 ± 0.6 ^d^	18.5 ± 0.9 ^d^	20.6 ± 0.3 ^c^	17.2 ± 0.3 ^d^	21.2 ± 0.3 ^c^	29.5 ± 0.3 ^b^
*γ*-Tocopherol (µg/g)	0.48 ± 0.06 ^g^	3.44 ± 0.06 ^d^	1.23 ± 0.02 ^f^	1.27 ± 0.06 ^f^	0.93 ± 0.02 ^f^	0.25 ± 0.02 ^g^	1.26 ± 0.12 ^f^	2.20 ± 0.06 ^e^	0.78 ± 0.06 ^f^	3.09 ± 0.06 ^d^
*δ*-Tocopherol (µg/g)	0.21 ± 0.03 ^e^	0.35 ± 0.03 ^c^	0.23 ± 0.02 ^d^	0.23 ± 0.06 ^de^	0.22 ± 0.03 ^de^	0.25 ± 0.06 ^d^	0.27 ± 0.03 ^d^	0.11 ± 0.03 ^e^	0.76 ± 0.03 ^b^	0.19 ± 0.03 ^e^
**Total tocopherols** **(µg/g)**	22.2 ± 0.6 ^e^	34.0 ± 1.2 ^c^	18.0 ± 0.6 ^f^	26.2 ± 0.3 ^d^	15.7 ± 0.6 ^f^	19.0 ± 0.9 ^f^	22.1 ± 0.4 ^e^	19.5 ± 0.3 ^f^	22.7 ± 0.3 ^e^	32.8 ± 0.3 ^c^
GABA (mg/g)	3.9 ± 0.6 ^c^	4.7 ± 0.0 ^c^	11.8 ± 1.2 ^a^	6.1 ± 0.3 ^b^	6.2 ± 1.4 ^b^	6.4 ± 1.1 ^b^	12.3 ± 0.3 ^a^	4.0 ± 0.3 ^c^	3.7 ± 0.5 ^c^	2.8 ± 0.6 ^d^
**Total amino acids (** **mg** **/g)**	28.4 ± 4.3 ^c^	39.0 ± 1.0 ^ab^	41.7 ± 3.9 ^ab^	35.9 ± 1.2 ^b^	31.6 ± 7.1 ^c^	36.5 ± 5.3 ^b^	47.5 ± 3.3 ^a^	30.7 ± 3.0 ^c^	29.8 ± 3.5 ^c^	27.6 ± 6.4 ^c^
Vitamin C (mg AAE/g)	2.06 ± 0.18 ^bc^	1.86 ± 0.04 ^c^	1.89 ± 0.04 ^c^	2.90 ± 0.05 ^a^	2.26 ± 0.12 ^bc^	2.48 ± 0.05 ^b^	1.94 ± 0.05 ^c^	1.70 ± 0.04 ^c^	2.44 ± 0.05 ^b^	1.98 ± 0.05 ^c^
**Antioxidant Activity of Lipophilic Extracts**										
Total phenolics (μmol GAE/g)	17.2 ± 0.1 ^b^	18.4 ± 1.0 ^a^	13.3 ± 0.3 ^d^	13.4 ± 0.3 ^d^	18.8 ± 0.1 ^a^	18.3 ± 0.1 ^a^	12.8 ± 0.2 ^d^	18.8 ± 0.5 ^a^	17.7 ± 0.6 ^b^	18.5 ± 0.1 ^a^
Total flavonoids (μmol QE/g)	ND	ND	ND	ND	ND	ND	ND	ND	ND	ND
DPPH (μmol TE/g)	29.7 ± 3.7 ^a^	17.6 ± 1.8 ^b^	26.4 ± 1.5 ^a^	11.4 ± 0.8 ^c^	27.3 ± 3.5 ^a^	25.2 ± 2.3 ^a^	19.0 ± 1.0 ^b^	19.8 ± 1.7 ^b^	28.7 ± 2.0 ^a^	21.6 ± 1.9 ^b^
**Antioxidant Activity of Hydrophilic Extracts**										
Total phenolics (μmol GAE/g)	16.3 ± 1.6 ^c^	17.8 ± 0.4 ^c^	18.3 ± 0.1 ^bc^	15.6 ± 0.7 ^d^	14.1 ± 2.2 ^d^	18.3 ± 0.6 ^bc^	13.0 ± 0.3 ^d^	15.4 ± 0.8 ^d^	27.2 ± 1.3 ^a^	16.6 ± 0.4 ^c^
Total flavonoids (μmol QE/g)	6.1 ± 0.2 ^d^	7.1 ± 0.5 ^c^	6.1 ± 0.4 ^d^	5.5 ± 0.3 ^d^	5.6 ± 0.3 ^d^	6.5 ± 0.4 ^c^	6.0 ± 0.3 ^d^	6.6 ± 0.5 ^c^	7.7 ± 0.6 ^b^	7.2 ± 0.6 ^c^
DPPH (μmol TE/g)	34.3 ± 2.4 ^c^	33.2 ± 2.8 ^c^	32.7 ± 2.5 ^c^	36.9 ± 0.7 ^b^	27.4 ± 1.8 ^d^	38.9 ± 4.9 ^b^	37.2 ± 5.5 ^b^	28.2 ± 2.3 ^d^	37.3 ± 2.0 ^b^	34.5 ± 2.7 ^c^
**Items**	**Cultivars**									
	**Blackchoi**		**Blackchoco**	**Greengana**		**Norangdotori**	**Orange**	**Tamina**	**GC-9**	**GC-19**
*L**	50.1 ± 0.3 ^d^		59.0 ± 0.3 ^b^	70.3 ± 0.6 ^a^		77.1 ± 2.5 ^a^	55.1 ± 0.4 ^c^	46.8 ± 0.6 ^de^	74.2 ± 1.0 ^a^	72.6 ± 1.0 ^a^
*a**	18.8 ± 0.1 ^d^		3.8 ± 0.1 ^e^	−11.7 ± 0.7 ^g^		−3.3 ± 0.5 ^f^	11.6 ± 0.5 ^d^	33.9 ± 0.7 ^a^	1.0 ± 0.5 ^e^	0.8 ± 0.3 ^e^
*b**	26.1 ± 0.2 ^d^		27.4 ± 0.3 ^cd^	31.1 ± 2.4 ^c^		35.8 ± 0.5 ^b^	47.0 ± 0.9 ^a^	33.2 ± 2.1 ^bc^	41.8 ± 1.0 ^b^	38.1 ± 2.7 ^b^
Hue angle (°)	20.9 ± 0.1 ^f^		37.8 ± 0.1 ^d^	65.1 ± 1.1 ^a^		48.1 ± 1.0 ^b^	36.5 ± 0.5 ^d^	14.7 ± 0.8 ^h^	43.2 ± 0.3 ^c^	42.4 ± 0.5 ^c^
Lutein (µg/g)	8.9 ± 0.9 ^b^		26.2 ± 2.6 ^a^	5.7 ± 1.5 ^b^		5.6 ± 1.0 ^b^	4.4 ± 0.4 ^b^	2.9 ± 0.1 ^b^	5.5 ± 0.2 ^b^	4.2 ± 0.3 ^b^
*β*-Carotene (µg/g)	49.1 ± 3.8 ^a^		55.0 ± 1.4 ^a^	0.9 ± 0.3 ^e^		0.9 ± 0.3 ^e^	41.0 ± 5.5 ^b^	36.5 ± 1.8 ^b^	2.6 ± 0.3 ^e^	1.6 ± 0.2 ^e^
Lycopene (µg/g)	294.5 ± 15.8 ^c^		190.0 ± 24.9 ^e^	ND		ND	ND	246.4 ± 19.1 ^d^	ND	ND
Other carotenoids (µg/g)	172.3 ± 26.6 ^ab^		137.9 ± 9.9 ^bc^	0.8 ± 0.1 ^d^		0.9 ± 0.0 ^d^	25.8 ± 0.8 ^d^	155.3 ± 4.1 ^b^	1.8 ± 0.3 ^d^	0.8 ± 0.3 ^d^
**Total carotenoids (µg/g)**	524.8 ± 47.1 ^b^		409.1 ± 38.8 ^c^	7.4 ± 1.8 ^f^		7.3 ± 1.3 ^f^	71.2 ± 6.8 ^e^	441.1 ± 25.1 ^c^	9.9 ± 0.8 ^f^	6.6 ± 0.7 ^f^
Chlorophyll a (µg/g)	65.6 ± 3.7 ^ab^		79.8 ± 18.9 ^a^	50.5 ± 4.8 ^b^		ND	23.8 ± 0.0 ^c^	ND	8.3 ± 2.5 ^c^	6.4 ± 2.7 ^c^
Chlorophyll b (µg/g)	16.6 ± 1.1 ^c^		56.1 ± 1.6 ^a^	27.5 ± 3.8 ^b^		9.8 ± 1.3 ^d^	10.8 ± 0.2 ^d^	ND	6.4 ± 0.8 ^d^	5.9 ± 1.1 ^d^
*α*-Tocopherol (µg/g)	27.1 ± 0.6 ^b^		29.5 ± 0.6 ^b^	20.9 ± 0.3 ^c^		28.6 ± 0.3 ^b^	23.3 ± 0.3 ^c^	5.5 ± 0.3 ^e^	61.8 ± 0.3 ^a^	58.3 ± 0.9 ^a^
*γ*-Tocopherol (µg/g)	9.36 ± 0.12 ^a^		10.1 ± 0.1 ^a^	3.00 ± 0.06 ^d^		3.67 ± 0.12 ^d^	6.67 ± 0.06 ^b^	4.21 ± 0.27 ^c^	3.45 ± 0.02 ^d^	4.18 ± 0.02 ^c^
*δ*-Tocopherol (µg/g)	1.58 ± 0.02 ^a^		1.65 ± 0.02 ^a^	0.91 ± 0.02 ^b^		0.87 ± 0.03 ^b^	1.25 ± 0.03 ^a^	0.42 ± 0.03 ^c^	0.62 ± 0.02 ^b^	0.75 ± 0.03 ^b^
**Total tocopherols (µg/g)**	38.0 ± 0.7 ^b^		41.3 ± 0.6 ^b^	24.8 ± 0.3 ^d^		33.1 ± 0.4 ^c^	31.2 ± 0.3 ^c^	10.2 ± 0.6 ^g^	65.9 ± 0.3 ^a^	63.3 ± 0.9 ^a^
GABA (mg/g)	2.6 ± 0.5 ^d^		8.2 ± 0.4 ^b^	7.5 ± 0.4 ^b^		2.1 ± 0.2 ^d^	2.5 ± 0.2 ^d^	3.3 ± 0.0 ^c^	3.5 ± 0.0 ^c^	3.7 ± 0.2 ^c^
**Total amino acids (mg/g)**	29.6 ± 6.8 ^c^		38.0 ± 3.7 ^ab^	38.0 ± 0.4 ^ab^		24.9 ± 3.2 ^c^	29.5 ± 1.1 ^c^	36.2 ± 0.4 ^b^	31.5 ± 1.1 ^c^	35.8 ± 3.1 ^b^
Vitamin C (mg AAE/g)	1.98 ± 0.04 ^c^		2.19 ± 0.10 ^bc^	0.23 ± 0.04 ^e^		0.43 ± 0.03 ^e^	0.23 ± 0.03 ^e^	2.60 ± 0.05 ^b^	1.10 ± 0.04 ^d^	1.06 ± 0.04 ^d^
**Antioxidant Activity of Lipophilic Extracts**										
Total phenolics (μmol GAE/g)	18.9 ± 1.2 ^a^		17.9 ± 0.1 ^ab^	15.8 ± 0.4 ^c^		17.2 ± 0.3 ^b^	19.1 ± 0.3 ^a^	13.1 ± 0.2 ^d^	15.0 ± 0.2 ^c^	18.1 ± 0.1 ^ab^
Total flavonoids (μmol QE/g)	ND		ND	ND		ND	ND	ND	ND	ND
DPPH (μmol TE/g)	18.8 ± 1.4 ^b^		11.4 ± 2.2 ^c^	10.8 ± 0.7 ^c^		9.9 ± 1.2 ^c^	9.0 ± 1.7 ^c^	25.1 ± 0.7 ^a^	10.2 ± 0.2 ^c^	8.8 ± 0.6 ^c^
**Antioxidant Activity of Hydrophilic Extracts**										
Total phenolics (μmol GAE/g)	14.6 ± 0.7 ^d^		19.1 ± 0.1 ^b^	13.4 ± 0.1 ^d^		18.2 ± 0.9 ^bc^	16.8 ± 0.2 ^c^	20.3 ± 0.2 ^b^	18.1 ± 0.2 ^c^	21.4 ± 0.2 ^b^
Total flavonoids (μmol QE/g)	6.7 ± 0.6 ^c^		6.9 ± 0.5 ^c^	7.2 ± 0.3 ^c^		10.8 ± 0.8 ^a^	10.8 ± 1.3 ^a^	8.6 ± 0.6 ^b^	10.7 ± 0.8 ^a^	10.6 ± 0.8 ^a^
DPPH (μmol TE/g)	31.2 ± 1.0 ^d^		35.3 ± 3.9 ^c^	15.7 ± 2.3 ^f^		22.8 ± 0.9 ^e^	20.5 ± 0.9 ^e^	41.9 ± 5.0 ^a^	26.4 ± 1.0 ^e^	23.9 ± 1.0 ^e^

*L**, lightness; *a**, + red—green; *b**, + yellow—blue; AAE, ascorbic acid equivalent; GAE, gallic acid equivalent; QE, quercetin equivalent; TE, trolox equivalent. Hue angle was obtained from the formula, *h*° = tan^−1^ (*b**/*a**). Data are expressed as the mean (the average value of content for dry weight) and SD (the standard deviation value) of three independent experiments. Different letters in the same row mean significantly different by Duncan’s multiple range test (*p* < 0.05). ND, not detected.

**Table 3 molecules-27-08741-t003:** Pearson’s correlation coefficients from physicochemical data, the contents of antioxidant constituents, and antioxidant activities of ten regular tomato cultivars.

Traits	TA	*L**	*a**	*b**	*h*°	Lut	*β*-Car	Lyc	TCar	TToco	GABA	TAA	AAC	TPhe-L	DPPH-L	TPhe-H	TFla-H	DPPH-H
Brix	−0.23	0.08	−0.35	0.08	0.39	−0.11	−0.15	−0.45	−0.49	0.19	−0.64 *	−0.58	−0.08	0.65 *	−0.01	0.34	0.60	−0.18
TA		−0.28	−0.17	−0.59	−0.34	−0.22	−0.03	0.58	0.55	0.03	0.72 *	0.71 *	−0.31	−0.60	−0.04	0.07	0.03	0.26
*L**			−0.77 **	−0.08	0.71 *	0.17	0.32	−0.79 **	−0.78 **	0.83 **	−0.50	−0.23	−0.04	0.08	−0.59	−0.07	0.24	−0.02
*a**				0.38	−0.67 *	−0.08	−0.29	0.64 *	0.67 *	−0.77 **	0.36	0.09	0.14	−0.02	0.56	−0.15	−0.38	−0.03
*b**					0.42	0.37	0.20	−0.02	0.03	−0.24	−0.41	−0.38	0.15	0.51	0.46	0.50	0.37	−0.05
*h* ^°^						0.37	0.47	−0.69 *	−0.68 *	0.56	−0.68 *	−0.39	−0.04	0.44	−0.21	0.49	0.62	−0.06

TA, total acids; *L**, lightness; *a**, + red—green; *b**, + yellow—blue; *h*°, Hue angle; Lut, lutein content; *β*-Car, *β*-carotene content; Lyc, lycopene content; TCar, total carotenoid content; TTC, total tocopherol content; GABA, *γ*-aminobutyric acid content; TAA, total amino acid content; AAC, ascorbic acid content; TPhe-L, total phenolic content in lipophilic extract; DPPH-L, antioxidant activity of lipophilic extract on DPPH assay; TPhe-H, total phenolic content in hydrophilic extract; TFla-H, total flavonoid content in hydrophilic extract; DPPH-H, antioxidant activity of hydrophilic extract on DPPH assay. * Significant at *p* < 0.05. ** Significant at *p* < 0.01.

**Table 4 molecules-27-08741-t004:** Pearson’s correlation coefficients from physicochemical data, the contents of antioxidant constituents, and antioxidant activities of six medium-sized tomato cultivars.

Traits	TA	*L**	*a**	*b**	*h*°	Lut	*β*-Car	Lyc	TCar	TToco	GABA	TAA	AAC	TPhe-L	DPPH-L	TPhe-H	TFla-H	DPPH-H
Brix	−0.36	−0.59	0.57	0.40	−0.62	−0.27	0.54	0.26	0.29	0.18	−0.73	−0.06	0.07	0.52	0.17	−0.04	0.29	0.11
TA		0.16	−0.41	−0.35	0.27	0.96 **	0.36	0.04	0.14	0.66	0.76	−0.45	0.15	0.37	−0.46	0.10	−0.42	0.07
*L**			−0.89 *	0.07	0.87 *	−0.01	−0.82 *	−0.76	−0.79	0.25	0.19	−0.54	−0.70	0.09	−0.71	−0.24	0.29	−0.69
*a**				0.01	−0.96 **	−0.20	0.62	0.73	0.73	−0.44	−0.47	0.74	0.72	−0.30	0.84 *	0.49	−0.01	0.79
*b**					0.10	−0.50	−0.16	−0.65	−0.60	−0.21	−0.46	−0.03	−0.60	0.16	−0.38	0.13	0.87	−0.42
*h*°						0.03	−0.72	−0.81 *	−0.82 *	0.21	0.48	−0.59	−0.78	0.12	−0.79	−0.50	0.08	−0.83 *

TA, total acids; *L**, lightness; *a**, + red—green; *b**, + yellow—blue; *h*°, Hue angle; Lut, lutein content; *β*-Car, *β*-carotene content; Lyc, lycopene content; TCar, total carotenoid content; TTC, total tocopherol content; GABA, *γ*-aminobutyric acid content; TAA, total amino acid content; AAC, ascorbic acid content; TPhe-L, total phenolic content in lipophilic extract; DPPH-L, antioxidant activity of lipophilic extract on DPPH assay; TPhe-H, total phenolic content in hydrophilic extract; TFla-H, total flavonoid content in hydrophilic extract; DPPH-H, antioxidant activity of hydrophilic extract on DPPH assay. * Significant at *p* < 0.05. ** Significant at *p* < 0.01.

**Table 5 molecules-27-08741-t005:** Pearson’s correlation coefficients of the contents of antioxidant constituents and antioxidant activities of tomato pulp with skin from 18 cultivars.

Traits	*L**	*a**	*b**	*h*°	Lut	*β*-Car	Lyc	TCar	TToco	GABA	TAA	AAC	TPhe-L	DPPH-L	TPhe-H	TFla-H	DPPH-H
TA	−0.27	0.05	−0.57 *	−0.15	0.65 **	0.40	0.44	0.46	−0.19	0.71 **	−0.28	0.21	−0.09	0.10	−0.02	−0.48 *	0.27
*L**		−0.83 **	0.33	0.78 **	−0.03	−0.60 **	−0.82 **	−0.83 **	0.69 **	−0.32	−0.30	−0.61 **	0.05	−0.74 **	0.03	0.57 *	−0.59 *
*a**			−0.33	−0.96 **	−0.18	0.48 *	0.84 **	0.85 **	−0.66 **	0.18	0.31	0.78 **	−0.13	0.83 **	0.01	−0.59 *	0.75 **
*b**				0.46	−0.26	−0.27	−0.61 **	−0.62 **	0.37	−0.50 *	0.03	−0.55 *	0.25	−0.35	0.35	0.80 **	−0.48 *
*h* ^°^					0.10	−0.51 *	−0.85 **	−0.87 **	0.54 *	−0.20	−0.21	−0.81 **	0.13	−0.76 **	0.01	0.61 **	−0.79 **
Lut						0.46	0.10	0.18	0.14	0.25	−0.20	0.22	0.15	−0.11	0.09	−0.25	0.19
*β*-Car							0.45	0.56 *	−0.32	0.04	0.15	0.45	0.28	0.39	−0.03	−0.42	0.46
Lyc								0.99 **	−0.61 **	0.52 *	0.10	0.78 **	−0.25	0.81 **	0.01	−0.77 **	0.76 **
TCar									−0.62 **	0.48 *	0.11	0.81 **	−0.18	0.81 **	0.00	−0.79 **	0.79 **
TToco										−0.29	−0.34	−0.40	0.19	−0.69 **	0.17	0.61 **	−0.38
GABA											−0.10	0.21	−0.55 *	0.15	−0.28	−0.55 *	0.21
TAA												0.28	−0.45	0.22	0.20	0.10	0.40
AAC													−0.20	0.64 **	0.17	−0.68 **	0.91 **
TPhe-L														−0.03	0.09	0.14	−0.31
DPPH-L															0.13	−0.60 **	0.60 **
TPhe-H																0.38	0.28
TFla-H																	−0.50 *

TA, total acids; *L**, lightness; *a**, + red—green; *b**, + yellow—blue; *h*°, Hue angle; Lut, lutein content; *β*-Car, *β*-carotene content; Lyc, lycopene content; TCar, total carotenoid content; TTC, total tocopherol content; GABA, *γ*-aminobutyric acid content; TAA, total amino acid content; AAC, ascorbic acid content; TPhe-L, total phenolic content in lipophilic extract; DPPH-L, antioxidant activity of lipophilic extract on DPPH assay; TPhe-H, total phenolic content in hydrophilic extract; TFla-H, total flavonoid content in hydrophilic extract; DPPH-H, antioxidant activity of hydrophilic extract on DPPH assay. * Significant at *p* < 0.05. ** Significant at *p* < 0.01.

## Data Availability

Not applicable.

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
