# Peer review of "Antioxidant Constituents and Activities of the Pulp with Skin of Korean Tomato Cultivars"

_molecules, 2022, doi:10.3390/molecules27248741_

Round 1

Reviewer 1 Report

The manuscript entitled “Antioxidant Constituents and Activities of the Pulp with Skin of Korean Tomato Cultivars” presented the antioxidant activity of 18 different cultivars of tomato in South Korea, they show most of the content present in these cultivars with different tests. The manuscript is presented well but it has some deficiencies that need to be addressed.

1)      There is not any correlation between firmness and other parameters, i.e., by looking at the firmness with TSS, the correlation is quite confusing although both should be correlated with each other, by comparing all the values with each other. You need to elaborate well in the discussion portion.

2)      You have presented only one chromatogram, although the story is about all the cultivars, you need to present all the chromatograms in supplementary portions.

3)      As mentioned in section 2.4, Flavonoids are phenolics known to have antioxidant activities significantly contributing to the health benefits of tomatoes. Therefore, the total phenolic and flavonoid content can indirectly confirm antioxidant activity. By looking at the antioxidant activity of lipophilic extracts the correlation is quite different.

4)      Minors,

 a) Correct the wording line 470, Korean to Korea or South Korea,

b) line 480, “fomato” to tomato

c) line 481, “classiciation” to classification.

d) line 482-481, the word after has been repeated, arrange it well.

Author Response

Thank you very much for your careful and thorough reading of this manuscript and for the thoughtful comments and constructive suggestions, which helps to improve the quality of this manuscript. We have considered your comments (Original Reviewer’s Comments, ORC) very closely in revising this manuscript as follows.

Reviewer 2 Report

My comments are in the attached PDF.
Authors should put the method before the Results and Discussion. 

Author Response

Thank you very much for your consideration. We have considered your comments (Original Reviewer’s Comments, ORC) in the annotated pdf file very closely in revising this manuscript as follows.

Round 2

Reviewer 1 Report

The authors have presented well all the suggestions, the paper can be accepted.